# How Effective Is the Green Development Policy of China’s Yangtze River Economic Belt? A Quantitative Evaluation Based on the PMC-Index Model

**DOI:** 10.3390/ijerph18147676

**Published:** 2021-07-19

**Authors:** Shengli Dai, Weimin Zhang, Jiamin Zong, Yingying Wang, Ge Wang

**Affiliations:** School of Public Administration, Central China Normal University, Wuhan 430079, China; daishengli@mail.ccnu.edu.cn (S.D.); zhangweimin@mails.ccnu.edu.cn (W.Z.); jiaminzong@mails.ccnu.edu.cn (J.Z.); 311719020202@home.hpu.edu.cn (Y.W.)

**Keywords:** Yangtze River Economic Belt, green development, policy text, PMC-Index model, quantitative evaluation

## Abstract

Although many countries around the world, especially China, highlight the strategy of green development, there has been little research evaluating the effectiveness of green development policies in local area. This study explores 16 policy texts with the theme of green development in the Yangtze River Economic Belt in China. Using the Policy Modeling Consistency Index (PMC-Index) model, the paper establishes a multi-input–output policy table and scientifically and systematically evaluates these policies. The results show that the average PMC index of the 16 policy texts is 6.83, indicating a high overall quality of policy texts. The index identifies two states of policy effectiveness as being good and excellent; 50% of the total texts fall into these categories and do not fall into the category of having a low level of policy effectiveness. Five indicators, including policy timeliness, social benefits, policy audience scope, and incentives and constraints, significantly impact the PMC-Index of the policy. Six representative policy samples were selected and analyzed. The advantages and disadvantages of the policy can be more fully understood by the degree of depression of the PMC’s three-dimensional curved surface (PMC-Surface) model. Finally, the paper provides theoretical recommendations for the optimization of the green development policies.

## 1. Introduction

China is the most populous developing country in the world, and its extensive mode of economic development hides high environmental costs that need to be addressed. To alleviate severe damage to the ecological environment and effectively address challenges, the Chinese government has proposed a new economic development model, called green development. This is a basic state policy that is closely linked to China’s modernization drive, and it could create significant benefits for the present and the future.

In addressing the contradiction between economic development and ecological environmental governance, China has progressed through three spiral stages: end-of-environment pollution control, sustainable development, and green development [1]. To ensure the further smooth progress of ecological and environmental protection, the Chinese government agencies and departments have been working on ambitious development plans with set milestones [2]. In 2011, the Chinese government established green development as the theme of the 12th Five-Year Plan. In 2016, the 13th Five-Year Plan noted that China’s development should always adhere to the concept of ecological priority and green development, and strengthen regional coordination and comprehensive management [3]. In 2021, the 14th Five-Year Plan outlines the need to adhere to green development roads, build the foundation for an ecological civilization, consolidate development advantages, solve development problems, and jointly promote ecological environment protection and economic development. The effect of environmental protection in China continues to be visible, and with its significant economic achievements, China is paying more attention to transforming from a traditional development model to a new green development mode [4].

As a major national strategic development area in China, the Yangtze River Economic Belt covers 11 provinces and cities, and makes up for 21.4% of China’s total area [5]. It is an important ecological barrier area in the Yangtze River Basin and in China. In recent years, the total economic output of the Yangtze River Economic Belt has reached at 45.78 trillion yuan ($6.92 trillion), contributing 49.7% of the total growth of China’s GDP [6]. The Yangtze Economic Belt is currently experiencing resource-related, environmental, and ecological problems. The main performance challenges include the extensive and wasteful use of resources, increasingly serious environmental pollution, and significant ecosystem degradation. These challenges have become a major bottleneck restricting the sustainable development of the area’s economy and society. In a meeting on the development of the Yangtze River Economic Belt held in early 2016, President Xi Jinping stressed “green development” of the region and said that restoring the ecological environment would be an overwhelming task. The support of national leaders has provided unprecedented opportunities for the green development of the Yangtze Economic Belt. Since then, the Chinese government has issued many policies and regulations to ensure that the economic development of the Yangtze River Economic Belt is combined with green development. For example, since the implementation of the Afforestation Plan and Fishing Ban in the Yangtze River Economic Belt, China has planted 880,000 hectares of trees along both sides of Yangtze River, and imposed a 10-year fishing moratorium in 332 conversation areas in the Yangtze River basin in 2020.

Policy evaluation serves as a very important part of the public policy analysis process [7]. It also serves as the basis for the rational allocation of policy resources, and effectively test the effects of policies. In the past, the government used pilot programs as a key way to modernize public services, and expected evaluators to act as change agents [8]. However, evidence highlights the limitations of this approach; instead, selecting a method for quantifying policy effects can achieve more effective and interactive governance [9]. China’s green development policy still lacks an accurate, scientific, and concise evaluation method to explore the advantages and disadvantages of these policies [10]. What are the shortcomings of the green development policies in practice? What should be the next steps for improvement? These questions need further exploration. By evaluating the green development policy of the Yangtze River Economic Belt, we can identify problems more accurately in a policy text. We can also summarize experiences and lessons, and provide a reference for a more optimal policy path.

The policy modeling research consistency index (PMC-Index) is a single index that can evaluate the strong or weak points of a policy modeling research paper [11]. The index provides a new way of thinking and a new perspective for the quantitative evaluation of policies. The PMC-Index model has no limit on the number of indicators, allowing it to comprehensively accommodate more variables in the evaluation [12]. In addition, the PMC-Index model can intuitively show the advantages and disadvantages of a single policy, as well as the score composition of each variable; this can support improvements in specific aspects of a policy [13]. In practice, the PMC-Index model has been consistently validated by several studies in the field of policy evaluation [11,14,15,16].

This research report proceeds as follows. First, we collect the policy texts related to this study, screen and identify the content of policy texts using text mining technology, and remove irrelevant samples. Second, based on the study’s theme and content, we design the first-level variables, and extend second-level variables that meet the research needs, and build an evaluation system. Third, we apply the Policy PMC-Index model as the main research method and use the binary counting method to calculate the second-level variables of policy texts. Next, according to the parameters of the policy samples, we establish a multi-input–output policy table. Finally, by constructing a matrix with the dimension 3*3 and drawing a PMC-Surface chart, the study evaluates the effectiveness, advantages, and disadvantages of the policy texts, examines the problems in each policy text, and provides specific recommendations.

## 2. Literature Review

China’s understanding of green development has come later than western countries [17]. In 2012, the United Nations Conference on Sustainable Development considered green transformation as a core topic, advancing the process of global green development [18]. Since then, green development has been a priority on China’s policy agenda [19]. Previous research on green development has focused on the concept of green development [20,21], green development efficiency [22,23,24], green development strategy [25,26,27], and green development level [28,29]. In recent years, scholars have deeply integrated green development with economic ecology and social ecology, and have conducted research from the aspects of green finance [30], green economy [31], green credit [32], and green energy [33]. Research on the evaluation of green development has focused on evaluating green development indexes [34], evaluating the performance of green development [35], and comprehensively evaluating green development [36].

In terms of the evaluation of green development policies, some scholars have explored the impact of policies on different fields. For example, Chen et al. [37] and Tang et al. [38] discussed the impact of green policy implementation on industrial structures and the construction industry. Shade et al. studied the implementation of green infrastructure policy and predicted the impact of the policy on future urban climate change response capacity [39]. A few scholars have applied a qualitative perspective. Realizing that China’s green innovation policy started late, Wu et al. [40] conducted in-depth research of developed countries to provide information that can improve green development policies. Others have evaluated these policies using quantitative methods. Liu et al. [41] studied evolutionary and development trends by establishing a policy evaluation system to facilitate a new understanding of green development policy in the construction industry. However, few studies have scientifically and systematically evaluated green development policies in China.

Policy evaluation is a key element of policy operations, can directly facilitate policy formulation and implementation, and provides an important reference value for adjusting policy [42]. Early research on policy evaluation focused on semantic and grammatical research. Scholars focused on conceptual interpretations and conducted comparative analyses of high-frequency words and subject words in policy texts [43]. In contrast, the mainstream paradigm of pragmatic research is the CDA (Critical Discourse Analysis) framework, which establishes the connection between policy texts and social reality, using descriptive language to infer the ideology underlying the policy text [44]. Most of these policy evaluations involve text interpretation [18]. Compared with early policy evaluations, later policy evaluations were improved with respect to the evaluation content, scope, standard, and method; they were more comprehensive, scientific and normative. However, most evaluation content remains in the policy text itself and is subjective [45].

As public policy research has deepened, the scope of policy evaluation has become more comprehensive and scientific and has gradually evolved from qualitative research to a comprehensive combination of qualitative and quantitative research [46]. By combining the advantages of qualitative research and quantitative research, combined research methods allow for the comprehensive and objective evaluation of policy text content. This also facilitates the development of empirical research on policy texts, which is significant in opening the black box of decision making [47].

In 2010, the PMC-Index model was proposed by Ruiz Estrada based on Cartesian space application and the Omnia Mobilis (everything is moving) hypothesis [48], which innovated a single policy evaluation method [49]. This method judges the consistency of the policy from a multidimensional perspective and allows for the direct observation of the advantages and disadvantages of the policy text by constructing a PMC-Surface [50]. In recent years, the application of the PMC-Index model has attracted significant attention in the academic community and has become a popular way to evaluate a policy’s effectiveness. For example, Zhang et al. [51] applied the PMC-Index model to policy studies about the construction industry, to evaluate the effectiveness and efficiency of research strategies. Li et al. [15] used the PMC-Index model to quantitatively evaluate policies about China’s pork industry. Scholars Kuang [52] and Peng [16] verified that the PMC-Index model could be applied to study China’s long-term care insurance policy and farmland protection policy, showing strong practical application value.

These studies highlight that the PMC-Index model could be useful for quantitatively evaluating the Yangtze River Economic Belt’s green development policy. Compared with existing comprehensive evaluation methods, the PMC-Index model is more targeted and operable, and can generally avoid subjectivity in the policy evaluation process, improving the accuracy of policy evaluation. Based on this, this study combined the PMC-Index model and an existing quantitative evaluation method to evaluate the effectiveness of green development policies in the Yangtze River Economic Belt. The goal was to fill this gap in the literature and provide a theoretical reference for policy optimization and innovation.

## 3. Research Design

### 3.1. Data Sources

This study analyzes policy texts based on national-level policies. The two channels considered include policy texts collected through relevant official websites and bulletins of the national ministries and commissions; and relevant information identified using the intelligent law retrieval database launched by the Peking University Law Department. Before collecting relevant texts, it is important to determine the subject and content of the policy. The core theme of the policy text relates to the green development of the Yangtze River Economic Belt. The main body of the text is limited to the State Council and national ministries and commissions and does not involve policy from government departments or industry standards. Based on this, 110 policy texts were reviewed and reviewed, including regulations, decisions, resolutions, opinions, rules, plans, plans, measures, detailed rules, and circulars. To focus only on texts with substantive policy content and concrete action plans, the final sample includes 16 policy texts. They are identified and coded as P1 to P16, as shown in Table 1.

### 3.2. PMC-Index Model

Using the 16 policy texts with the theme of green development in the Yangtze River Economic Belt, this study conducts a quantitative policy evaluation by applying the calculation steps of the PMC-Index model. These steps are as follows:Apply Equations (1) and (2) to design the first-level variables. Then, the second-level variables are extended in a standardized form. The second-level variables are derived from first-level variables, and the sum of the ratios of the scores of the second-level variables to the number of second-level variables represents the value of the first-level variables. All the variables involved are aligned to fit a [0, 1] distribution.Assign PMC-Index values. Using content from the policy texts, the second-level variables identified in Step 1 are assigned using the binary counting method. If the content related to the second-level variable appears in the policy text, it is coded as a 1, otherwise, it is 0. Then, the multi-input–output policy table is constructed.Calculate the PMC-Index. Using Step 2, the scores of the second-level variables are added up under each first-level variable. The algorithm is shown in Equation (3). In Equation (3), it is the ordinal number of the first-level variable; j is the ordinal number of the second-level variable; X_t_ is the t-th first-level variable; X_tj_ is the score of the second-level variable; T(X_tj_) is the number of the second-level variable under the corresponding first-level variable.According to Equation (4), the PMC-Surface charts are constructed.
(1)$$X\sim N\;\left[0,1\right],$$
(2)$$X=\left\{\mathrm{XR}:\left[0\sim 1\right]\right\},$$
(3)$$X_{t}(\overset{n}{\underset{j=1}{\sum }}\frac{X_{\mathrm{tj}}}{T\left(X_{\mathrm{tj}}\right)})t=1,\;2,\;3\cdots 8,\;9,$$
(4)$$\mathrm{PMC}=\left[X_{1}\left(\sum_{i=1}^{4}\frac{X_{1i}}{4}\right)+X_{2}\left(\sum_{j=1}^{4}\frac{X_{2j}}{4}\right)+X_{3}\left(\sum_{k=1}^{3}\frac{X_{3k}}{3}\right)+X_{4}\left(\sum_{l=1}^{5}\frac{X_{4l}}{5}\right)+\;X_{5}\left(\sum_{m=1}^{6}\frac{X_{5m}}{6}\right)+X_{6}\left(\sum_{n=1}^{3}\frac{X_{6n}}{3}\right)+X_{7}\left(\sum_{o=1}^{2}\frac{X_{7o}}{2}\right)+X_{8}\left(\sum_{p=1}^{6}\frac{X_{8p}}{6}\right)+\;X_{9}\left(\sum_{q=1}^{8}\frac{X_{9q}}{8}\right)+X_{10}\right]$$

### 3.3. Variable Design

Based on a review of the 16 policy texts concerning green development in the Yangtze River Economic Belt, and based on Ruiz Estrada [53] and previous policy evaluation studies, this study identifies 10 first-level variables and 41 s-level variables. The first-level variables, listed in Table 2, are numbered X1 to X10, including the nature of the policy and its function, time frame of policy effectiveness, policy evaluation and social benefits, policy subject and object, and implementation guarantees with respect to process and results.

Among the 10 first-level variables, the nature of the policy is used to determine whether the policy to be evaluated has the characteristics of oversight, prediction, recommendations, and experimentation. Policy function is used to determine whether the policy has the functions of normative guidance, classified oversight, collaborative management, and overall coordination. Policy timeliness is used to estimate whether the effectiveness of the policy is expected to be long-term (more than 5 years), medium-term (3–5 years), or short-term (less than 3 years). Content evaluation focuses on the following five aspects: whether the policy making is aligned with the problem and is based on sufficient basis, whether the policy planning is detailed, whether the policy program is scientific and reasonable, whether the goal of the policy is specific, and whether the policy has customized regional characteristics.

Social benefits are used to evaluate whether the policy can bring six benefits to societal development, including environmental protection, green development, circular economy, sustainability, a sound mechanism, and win–win cooperation. The policy subjects and objects are used to determine whether the policy content involves the subjects and objects listed in the table. Incentives, constraints, and implementation guarantee are used to determine whether the policy involves economic incentives (e.g., pollution charges, emissions trading, ecological compensation systems), tax concessions, financial subsidies, convenient services, administrative penalties, capital investments, assessments, publicity, and guidance, self-regulation, government regulation, law rules, policy support, social oversight, or technological innovation. Policy disclosure is used to determine whether the policy is open and transparent.

Using variable standardization at all levels, the content analysis method and binary counting method are applied to complete calculations for the 16 selected policy texts. If the policy text conforms to one of the second-level variables, it is coded as 1; otherwise, it is coded as 0. Table 3 shows the specific evaluation criteria for the second-level variables.

Different parameter variables have different meanings in the policy text; however, they are closely related. The analytical framework shown in Figure 1 was designed to assess the feasibility of policy evaluation tools and the relationship between different variables. In advancing the green development of the Yangtze River Economic Belt, the effectiveness and applicability of government policies will be affected by policy nature, policy timeliness, policy subjects, and policy objects. Policy design strategies and paths also differ because each policy applies to different fields and scenarios. Therefore, before evaluating policy implementation results, it is important to evaluate the basis of the problem, content planning and scheme, and goals and characteristics of the policy content. It is then important to select appropriate policy tools as incentives, restraints, and guarantees to prevent “policy friction”. By reasonably combining policies, we can achieve the expected policy goals and social benefits and support policy implementation.

### 3.4. PMC-Surface Construction

The PMC-Surface visualizes the PMC-Index in the form of a stereo image, which can visually display the advantages and disadvantages of the policy [50]. The establishment of the PMC-Matrix is the basis of PMC-Surface structure. Because all policy texts have the same score of 1 with respect to open and transparent policy disclosure (X10), considering the symmetry and balance requirements of the matrix, this variable is removed to standardize the matrix, showing the operation of the second-level variables. A matrix with the dimension of 3*3 is used to apply the PMC-Index with the remaining 9 variables to create the surface drawing. The associated matrix is shown in Equation (5).
(5)$$\mathrm{PMC}=\left[\begin{array}{ccc} X_{1} & X_{2} & X_{3}\\ X_{4} & X_{5} & X_{6}\\ X_{7} & X_{8} & X_{9} \end{array}\right]$$

## 4. Results and Discussion

### 4.1. Multi-Input–Output Tables

A multi-input–output table is designed to quantify the variable values. The content analysis and binary counting method are used to unify variable standards at all levels, yielding input–output values for 16 policy texts on green development in the Yangtze River Economic Belt. Based on Table 3, if the policy text conforms to one of the second-level variables, it is coded as 1; otherwise, it is coded as 0. As one example, policy P1 involves technological innovation (X9.8); as such, X9.8 of policy P1 should be coded as 1 in Table 4. Using the PMC-Index algorithm in Equation (3), the PMC-Indexes of 16 policies are summarized in Table 5. The policies are divided into four types to generate the PMC-Index hierarchy: a PMC-Index ranging from 0 to 4.99 indicates a low policy effectiveness, with needs for improvement; a PMC-Index ranging from 5 to 6.99 indicates a generally acceptable policy effectiveness. Scores ranging from 7 to 8.99 indicate excellent policy effectiveness; scores ranging from 9 to 10 indicate perfect policy effectiveness.

Table 5 shows that the PMC-Indexes of the 16 policies on the green development of the Yangtze River Economic Belt fluctuate between 5 and 8, showing that policy texts were classified as either having general or excellent effectiveness. The policy texts with general effectiveness are P3, P4, P5, P6, P9, P13, P15, and P16, with PMC-Indexes of 6.48, 6.33, 6.88, 5.50, 6.92, 5.29, 6.75, and 6.50, respectively. The policy texts with excellent effectiveness are P1, P2, P7, P8, P10, P11, P12, and P14, and their PMC-Index is 7.04, 7.38, 7.13, 7.08, 7.79, 7.83, 7.13, and 7.29.

Of these, P11 refers to the “Notice on Issuing the Three-Year Action Plan for Further Development of Multimodal Transport in the Yangtze River Economic Belt”. It was top ranked with respect to effectiveness, with the highest PMC-Index of 7.83. P13 is the Notice on Issuing the “Implementation Plan of the Central Finance Promotion Policy for Ecological Protection and Restoration of the Yangtze River Economic Belt”. It had the lowest PMC-Index of the 16 texts, at a score of 5.29. This delineated the policy as general and acceptable.

Of the 10 s-level variables, the 16 policy texts are all open and transparent, giving all a score of 1 for policy disclosure (X10). When X10 is excluded, the highest variable score is the policy content evaluation represented by content evaluation (X4), at 0.96. Only the X4 scores for policy P3 and policy P12 are lower than the average. The lowest score is policy timeliness (X3), at a score of 0.46.

This leads to the conclusion that in the policy-making process, policymakers follow a rigorous and scientific way of thinking, and focus on improving the resilience and feasibility of policies. The relevant departments also tend to formulate policy texts based on unified timeliness, setting a limited implementation period for participants.

### 4.2. Comparison of 6 Green Development Policies

Due to limited space, and given that all 16 policies were considered to have general and excellent effectiveness, to better distinguish the 16 policies, the secondary evaluation grades are classified according to the evaluation scores of policy effectiveness. A score between 7 and 7.99 indicates excellent policy effectiveness; a score between 6 and 6.99 indicates good policy effectiveness; a score between 5 and 5.99 indicates general policy effectiveness that would benefit from improvements. Based on these scores, maximum and minimum PMC-Indexes are selected from policies: P11, P1, P9, P4, P6, and P13, which have excellent, good, and general policy effectiveness. These six samples are used to construct the PMC-Surface charts, as shown in Figure 2, Figure 3, Figure 4, Figure 5, Figure 6 and Figure 7.

In the figures, the convex part of the surface indicates that the corresponding evaluation index scores are higher, and the concave part indicates that the corresponding evaluation index score is lower. The depression index is the total score of the first-level variable minus the PMC-Index. A larger depression index is associated with a larger degree of depression; as such, a lower first-variable value is expressed by having a lower position on the PMC-Surface in the three-dimensional model. The reverse is also true. The following section describes the six policies, combined with PMC-Surface charts and specific variable scores.

The PMC-Index of P11 is 7.83, ranking first among 16 policies. This specific policy is the “Notice on Issuing the Three-Year Action Plan for Further Development of Multimodal Transport in the Yangtze River Economic Belt” issued by the General Office of the Ministry of Transport of China. As such, it focuses on how to rely on river-combined transport to mitigate infrastructure shortcomings, and to implement green development policies in an all-around way by innovating intermodal service models and improving intermodal equipment. A review of the content of the policy indicates that its deficiency lies in the lack of supplemental social benefits (X5), which is related to the nature of the policy itself.

Policy text P1 is a notice issued by the National Development and Reform Commission of China on the Notice on Revising and Issuing “Special Management Measures for Central Budget Investment in Major Regional Development Strategies Construction (Green Development Direction of the Yangtze River Economic Belt)”. This policy text focuses on absorbing different investment entities, maximizing the role of the central government and increasing the efficiency of financial resource configuration. This shows a focus on solving the problems arising from the application of the project at this stage, but does not incorporate mid-term and long-term planning (X3). In the policy text, there are more descriptions of regulatory measures for the ecological environment and green development projects in the Yangtze River Economic Belt. In contrast, there were fewer references to incentives and constraints (X8) and implementation guarantees (X9) for all parties, leading to lower scores for these two items.

Policy texts with good policy effectiveness are exemplified by P9 and P4. According to the PMC-Surface charts of P9 and P4 in Figure 4 and Figure 5, the maximum and minimum values of the first-level variable scores of P9 and P4 are consistent with each other, at 1 and 0.33, respectively. Compared with P4, the average value of the first-level variable of P9 is 0.06 points higher, and the depression index is 3.08 and 3.67 for P9 and P4, respectively. The degree of depression in the surface is slightly deeper compared to policies P11 and P1.

From the perspective of policy content, P9 is the “Guiding Opinions on Strengthening Afforestation in the Yangtze River Economic Belt”. It focuses on describing how to build ecological green corridors through overall coordination and careful deployment to improve the optimal function of the forest ecosystem in the Yangtze River Economic Belt. However, the text does not mention how to gather all forces at a specific stage to jointly promote the green development of the Yangtze River Economic Belt, to realize a new pattern of win–win cooperation, and to share the fruits of sustainable development.

Policy text P4 is the “Guiding Opinions on Improving Sewage Treatment Charging Mechanism in Yangtze River Economic Belt”. The biggest difference between this policy and P9 is that the policy text adheres to the strategic positioning of ecological priority and green development, and gathers the joint forces of society, enterprises, and government departments to prevent and control water pollution. However, it has insufficient oversight in the policy implementation process, especially in the areas of assessment, self-regulation, social oversight, and technological innovation. As such, it cannot fully provide a policy guarantee. In summary, the above two policies do not fully reflect detailed planning of policy timeliness (X3), the conscious pursuit of social benefits (X5), and the specific requirements for policy objects (X6).

Policy texts with general policy effectiveness are exemplified by P6 and P13. Figure 6 and Figure 7 show that the PMC-Surface charts of P6 and P13 have the same maximum and minimum values of the first-level variable scores, which are 1 and 0.25, respectively. The average value of the first-level variable of P6 is 0.55, with a PMC-Index of 5.50. The average value of the first-level variable of P13 is 0.53, with a PMC-Index of 5.29, ranking lowest among the 16 policy texts. Although the average value and PMC-Index are similar, the degree in the fluctuation of each first-level variable differs.

From the perspective of policy content, P6 is the “Notice on Strengthening the Ecological Flow Oversight of Small Hydropower Stations in the Yangtze River Economic Belt”. The policy text has a clear theme, focusing on how to oversee the river basin storage and release work to ensure the ecological balance of the Yangtze River, maintain the health of the Yangtze River, and address the problem of ecological water use. Therefore, the nature of the policy (X1) is relatively thin and lacks the content of recommended measures, development forecasts, and pilot work.

Policy text P13 is the Notice on Issuing the “Implementation Plan of the Central Finance Promotion Policy for Ecological Protection and Restoration of the Yangtze River Economic Belt”, intended to advance the green development and ecological governance of the Yangtze River Basin and improve the comprehensive ecological service capacity of the Yangtze River Economic Belt. The policy function has not achieved real benefits, and it is inadequate in terms of overall coordination, classified oversight, and coordinated development. In addition to the above challenges, the two policies have low scores in terms of policy timeliness (X3), social benefits (X5), policy objects (X6), incentives and constraints (X8), and implementation guarantees (X9).

## 5. Conclusions

Green development means fostering economic growth and development while ensuring that natural assets continue to provide the resources and environmental services needed for our well-being [54]. To pursue the new vision of green development and a way of life and work that is green, low-carbon, circular and sustainable, governments around the world have actively implemented green development policies in recent years [55]. Green development policy plays an important guiding role in maintaining the balance between economic development and environmental protection, so the economy and society can achieve sustainable development [56].

The Yangtze River Economic Belt is an inland river economic belt with global influence and serves as a pioneering demonstration plot for the construction of an ecological civilization. Therefore, evaluating the green development policies of the region has important reference value for formulating regional green development policies around the globe. In response to this, this study focuses on 16 policy texts with the theme of green development in the Yangtze River Economic Belt and presents a quantitative analysis by building a PMC-Index model. We particularly focus on analyzing the advantages and disadvantages of these policies by selecting representative policies for PMC-Surface drawing. Finally, correspondingly targeted policy optimization paths for different types of green development policy are proposed.

This evaluation outcome indicates that the policy system for green development of the Yangtze River Economic Belt is relatively mature, with a reasonable overall design. The policymakers and issuers of each policy come from different departments. As such, the areas and themes of concern in each department differ, making them difficult to analyze. However, it is clear that the green development model has played a fundamental role in a specific period, effectively mitigating significant challenges facing the ecological environment. There are also, however, some weaknesses in the policies.

This study compares different points of emphasis with respect to the green development policy in China’s Yangtze River Economic Belt. Some policies direct the direction of green development at a macro level, and some define policy objectives at a micro perspective. A higher index score of the policy text was evaluated as “excellent.” The PMC-Index of policy P6 and policy P13 is relatively low at values of less than 6, highlighting opportunities for improvements. The internal division of some policies is clear and highlights a certain tendency. In particular, the five indicators of policy timeliness, social benefits, policy audience and scope, and incentives and constraints significantly impact the PMC-Index of the policy. The PMC-Surface charts created with six policy samples show the policy weaknesses, and allow for the visualization of an optimization path for each policy text. In general, although there is still room for improvement in some items, the relevant policies issued by the Chinese government have a relatively high quality and level in terms of completeness, rationality, and scientific foundations. This indicates they do contribute to the green development model.

Based on the evaluation of the PMC-Index and the degree of depression of the PMC-Surface, this study proposes corresponding paths for optimizing green development policy in China’s Yangtze River Economic Belt. The results show that the PMC-Surfaces of P11 and P1 exhibit excellent policy effectiveness evaluation grades and that they are located in the middle and upper part of the three-dimensional model. However, the two policies scored lower in terms of social benefits and safeguards in the implementation process. P11 has only one first-level variable with a below average score, indicating there is a slightly insufficient focus on short-term and specific planning. Policy should be based on overall interests, and effort is needed to unite the social interests of 11 provinces and cities in the Yangtze River Economic Belt; this should result in coordinated development and a new win–win situation. Three first-level variables in P1 are below the average value and may benefit from a more optimized path of X3-X9-X8.

In addition, the curved surface contours of P9 and P4 move down to the middle area of the three-dimensional model due to differences in scores. The variable P9 has 3 first-level variables with scores lower than the average, and P4 has 4. Therefore, it is recommended that the path to optimizing P9 is X3-X5-X6, and the optimization path for P4 is X3-X6-X5-X9. The two policies classified as having general effectiveness (P6 and P13) are located in the middle or even lower area of the three-dimensional model; and more than half of the first-level variable scores evaluated by the two policies are lower than the average. It highlights possible areas for improvement in the two policies. Therefore, the recommended optimal path for P6 should be improving X1-X3-X6-X8-X5-X9, and P13 should be improved in the variable path X2-X3-X9-X1-X5-X8. These cases are provided for their reference value, not as absolute directives.

This study emphasizes the advantages of traceability when using the PMC-Index model to evaluate the effectiveness of green development policies in the Yangtze River Economic Belt. If specific reasons for revisiting a policy are needed, it can be traced back to the input–output policy table (Table 4). This evaluation allows us to check the problems in a policy text from a comprehensive perspective, and provides lessons and a reference for policy optimization. In contrast to previous studies, this study specifically explores a single policy along different dimensions by establishing a 3D model of the PMC curved surface. It also compares the similarities and differences between different policies from a micro-perspective, providing a path to optimization. This is an advanced path in policy analysis. Scholars tend to mainly study green development policy at a macro-level, with little relevant discussion on the effect of the policy at the micro-level. Based on the restriction of conditions, this study reduces the possibility of subjective judgment by setting fixed-parameter variables as the research basis; however, some personal preferences are generally introduced in the process of text screening, variable design, and content recognition. Our findings also have highlighted several practical recommendations, which may lead to positive adjustments in the green development policy. Policy makers should first examine the integrity and consistency of a policy using a quantitative tool like PMC-Index Model. This can help identify the advantages and disadvantages of the policy. Then, the reliability and validity of a policy can be enhanced by increasing the key points of the policy, broadening the application fields of the policy, and rationally optimizing the distribution of attention in policy regulation.

In the future, a more detailed analysis can be made with respect to the accuracy of each parameter variable and the policy implementation effect. In addition to selecting universal indicators for evaluation, some non-universal indicators are established based on the characteristics of a specific policy. To reduce the subjectivity when selecting and setting evaluation indicators, we can combine the methods of bibliometric analysis, grounded theory coding, web crawler technology, and big data analysis to mine the characteristics of policy texts from more dimensions. In addition, green development represents an important trend both nationally and globally. As such, the scope of policy samples can be further expanded to national and global levels in the future.

## Figures and Tables

**Figure 1 ijerph-18-07676-f001:**
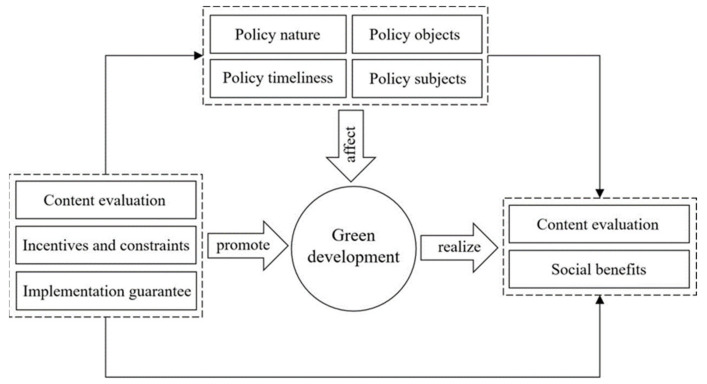
Relationships between different variable parameters.

**Figure 2 ijerph-18-07676-f002:**
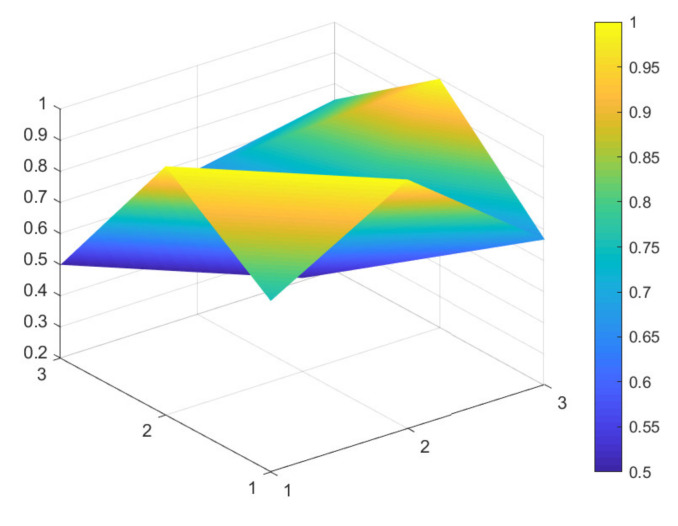
PMC-Surface chart of P11 (Excellent).

**Figure 3 ijerph-18-07676-f003:**
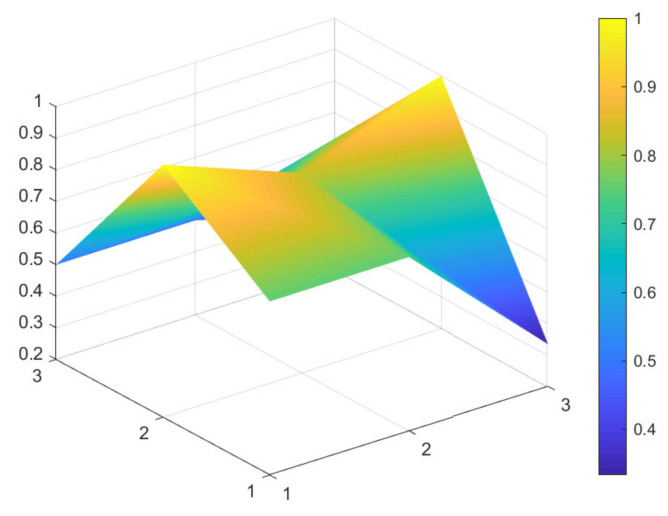
PMC-Surface chart of P1 (Excellent).

**Figure 4 ijerph-18-07676-f004:**
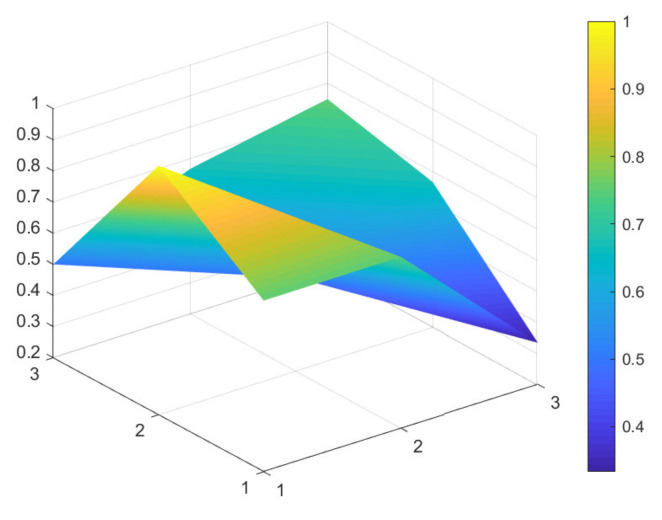
PMC-Surface chart of P9 (Good).

**Figure 5 ijerph-18-07676-f005:**
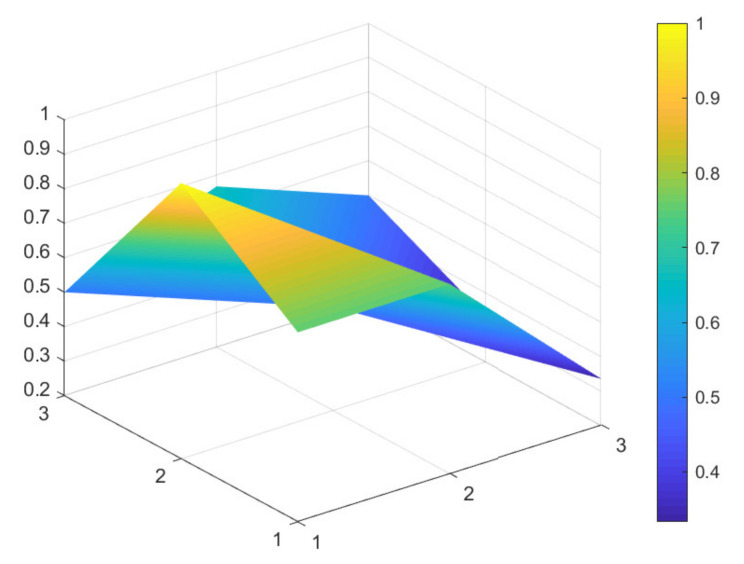
PMC-Surface chart of P4 (Good).

**Figure 6 ijerph-18-07676-f006:**
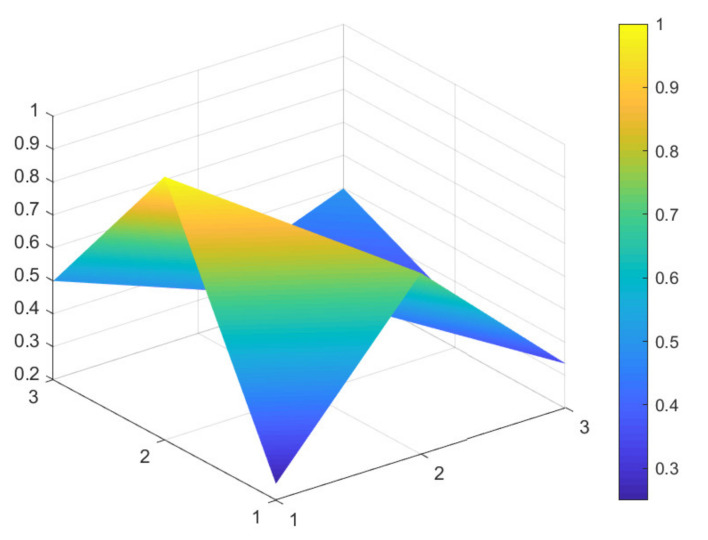
PMC-Surface chart of P6 (General).

**Figure 7 ijerph-18-07676-f007:**
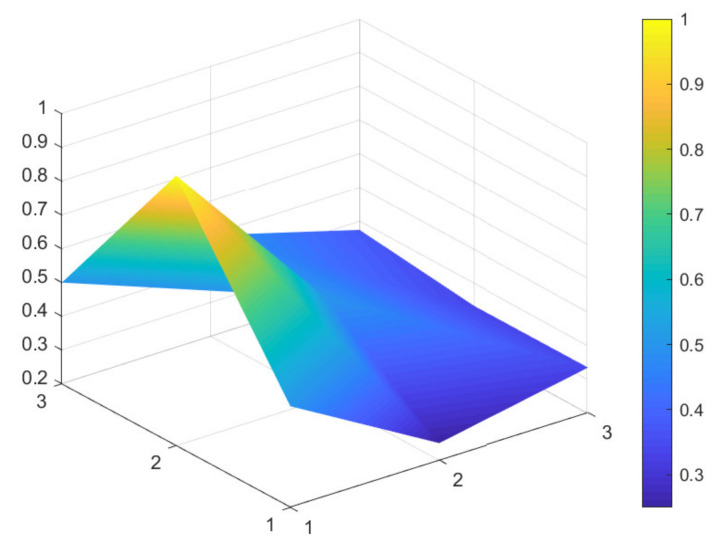
PMC-Surface chart of P13 (General).

**Table 1 ijerph-18-07676-t001:** Policy texts summary.

Code	Policy Text Name	Date
P1	Notice on Revising and Issuing “Special Management Measures for Central Budget Investment in Major Regional Development Strategies Construction (Green Development Direction of the Yangtze River Economic Belt)”	9 April 2021
P2	“Opinions on Establishing and Improving the Long-term Mechanism for Pollution Prevention and Control from Ships and Ports in the Yangtze River Economic Belt”	27 March 2021
P3	Notice on Issuing the “Implementation Plan for Strengthening Prevention and Control of Tailing Ponds Pollution in the Yangtze River Economic Belt”	26 February 2021
P4	“Guiding Opinions on Improving Sewage Treatment Charging Mechanisms in Yangtze River Economic Belt”	7 April 2020
P5	“Notice on Issuing the Remediation Plan for Prominent Pollution Issues from Ships and Ports in the Yangtze River Economic Belt”	17 January 2020
P6	“Notice on Strengthening the Ecological Flow Oversight of Small Hydropower Stations in the Yangtze River Economic Belt”	21 August 2019
P7	“Notice on Issuing the Key Work Points for Promoting Green Development of Agriculture and Rural Areas along the Yangtze River Economic Belt in 2019”	19 March 2019
P8	“Opinions on Carrying out the Rectification of Small Hydropower in the Yangtze River Economic Belt”	6 December 2018
P9	“Guiding Opinions on Strengthening Afforestation in the Yangtze River Economic Belt”	24 February 2016
P10	“Implementation Opinions on Supporting the Green Development of Agriculture and Rural Areas in the Yangtze River Economic Belt”	11 September 2018
P11	“Notice on Issuing the Three-Year Action Plan for Further Development of Multimodal Transport in the Yangtze River Economic Belt”	13 August 2018
P12	“Guiding Opinions on Establishing and Improving the Long-term Mechanism of Ecological Compensation and Protection in the Yangtze River Economic Belt”	13 February 2018
P13	Notice on Issuing the “Implementation Plan of the Central Finance Promotion Policy for Ecological Protection and Restoration of the Yangtze River Economic Belt”	30 January 2018
P14	“Guiding Opinions on Promoting Green Shipping Development in the Yangtze River Economic Belt”	4 August 2017
P15	“Guiding Opinions on Strengthening Green Industry Development in the Yangtze River Economic Belt”	30 June 2017
P16	“Guiding Opinions on Accelerating Postal Industry Development in the Yangtze River Economic Belt”	1 November 2016

Table 1 shows the names and release dates of 16 green development policy texts on the Yangtze River Economic Belt in China.

**Table 2 ijerph-18-07676-t002:** Policy variable design.

Number	First-Level Variables	Number	Second-Level Variables	Number	Second-Level Variables
X1	Policy nature	X1.1	Oversight	X1.2	Prediction
		X1.3	Recommendation	X1.4	Experimentation
X2	Policy function	X2.1	Normative guidance	X2.2	Classified oversight
		X2.3	Collaborative management	X2.4	Overall coordination
X3	Policy timeliness	X3.1	Short term (<3 years)	X3.2	Medium-term (3–5 years)
		X3.3	Long term (>5 years)		
X4	Content evaluation	X4.1	Sufficient basis	X4.2	Detailed planning
		X4.3	Scientific program	X4.4	Specific goals
		X4.5	Distinctive features		
X5	Social benefits	X5.1	Environmental protection	X5.2	Green development
		X5.3	Circular economy	X5.4	Sustainability
		X5.5	Sound mechanism	X5.6	Win-win cooperation
X6	Policy objects	X6.1	Industry	X6.2	Enterprise
		X6.3	Related departments		
X7	Policy subjects	X7.1	State Council	X7.2	National ministries
X8	Incentives and constraints	X8.1	Economic incentives	X8.2	Tax concessions
		X8.3	Financial subsidy	X8.4	Convenient service
		X8.5	Administrative penalty	X8.6	Capital investment
X9	Implementation guarantee	X9.1	Assessment	X9.2	Publicity and guidance
		X9.3	Self-regulation	X9.4	Government regulation
		X9.5	Law rules	X9.6	Policy support
		X9.7	Social oversight	X9.8	Technological innovation
X10	Policy disclosure				

Table 2 shows the basis of policy evaluation: 10 first-level variables and 41 second-level variables.

**Table 3 ijerph-18-07676-t003:** Evaluation criteria of second-level variables.

Number	Variable	Evaluation Criteria
X1	X1.1 Oversight	Whether the policy has oversight characteristics; if yes, it is 1; if no, it is 0
	X1.2 Prediction	Whether the policy is predictive; if yes, it is 1; if no, it is 0
	X1.3 Recommendation	Whether the policy has recommended content; if yes, it is 1; if no, it is 0
	X1.4 Experimentation	Whether the policy contains pilot demonstration projects; if yes, it is 1; if no, it is 0
X2	X2.1 Normative guidance	Whether the policy has the function of normative guidance; if yes, it is 1; if no, it is 0
	X2.2 Classified oversight	Whether the policy to be evaluated has the function of classified oversight; if yes, it is 1; if no, it is 0
	X2.3 Collaborative management	Whether the policy has the function of collaborative management; if yes, it is 1; if no, it is 0
	X2.4 Overall coordination	Whether the policy has the function of coordinating all forces; if yes, it is 1; if no, it is 0
X3	X3.1 Short term	Whether the policy involves short-term impact (terms <3 years); if yes, it is 1; if no, it is 0
	X3.2 Medium-term	Whether the policy involves medium-term impact (3–5 years); if yes, it is 1; if no, it is 0
	X3.3 Long term	Whether the policy involves long-term impact (terms >5 years); if yes, it is 1; if no, it is 0
X4	X4.1 Sufficient basis	Whether the problem basis of the policy is sufficient; if yes, it is 1; if no, it is 0
	X4.2 Detailed planning	Whether the content of the policy is detailed; if yes, it is 1; if no, it is 0
	X4.3 Scientific program	Whether the policy plan is scientific and reasonable; if yes, it is 1; if no, it is 0
	X4.4 Specific goals	Whether the goal of the policy is specific; if yes, it is 1; if no, it is 0
	X4.5 Distinctive features	Whether the policy has customized regional characteristics; if yes, it is 1; if no, it is 0
X5	X5.1 Environmental protection	Whether the policy has contributed to improving the efficiency of environmental governance; if yes, it is 1; if no, it is 0
	X5.2 Green development	Whether the policy contributes to green development; if yes, it is 1; if no, it is 0
	X5.3 Circular economy	Whether the policy attaches importance to circular economy development; if yes, it is 1; if no, it is 0
	X5.4 Sustainability	Whether the policy attaches importance to sustainable development; if yes, it is 1; if no, it is 0
	X5.5 Sound mechanism	Whether the policy has the utility of a sound mechanism; if yes, it is 1; if no, it is 0
	X5.6 Win-win cooperation	Whether the policy advocates collaborative governance and win-win cooperation; if yes, it is 1; if no, it is 0
X6	X6.1 Industry	Whether the policy object includes industry; if yes, it is 1; if no, it is 0
	X6.2 Enterprise	Whether the policy object includes enterprises; if yes, it is 1; if no, it is 0
	X6.3 Related departments	Whether the policy object includes related departments; if yes, it is 1; if no, it is 0
X7	X7.1 State Council	Whether the subject of the policy is the State Council; if yes, it is 1; if no, it is 0
	X7.2 National ministries	Whether the subject of the policy is a national ministry; if yes, it is 1; if no, it is 0
X8	X8.1 Economic incentives	Whether the policy contains measures for economic incentives; if yes, it is 1; if no, it is 0
	X8.2 Tax concessions	Whether the policy includes measures for tax concessions; if yes, it is 1; if no, it is 0
	X8.3 Financial subsidy	Whether the policy includes measures for financial subsidies; if yes, it is 1; if no, it is 0
	X8.4 Convenient service	Whether the policy involves content that facilitates services; if yes, it is 1; if no, it is 0
	X8.5 Administrative penalty	Whether the policy involves the administrative penalties; if yes, it is 1; if no, it is 0
	X8.6 Capital investment	Whether the policy involves the capital investment; if yes, it is 1; if no, it is 0
X9	X9.1 Assessment	Whether the policy involves assessment; if yes, it is 1; if no, it is 0
	X9.2 Publicity and guidance	Whether the policy involves publicity and guidance; if yes, it is 1; if no, it is 0
	X9.3 Self-regulation	Whether the policy involves self-regulation; if yes, it is 1; if no, it is 0
	X9.4 Government regulation	Whether the policy involves government oversight; if yes, it is 1; if no, it is 0
	X9.5 Law rules	Whether the policy involves legal rules; if yes, it is 1; if no, it is 0
	X9.6 Policy support	Whether the policy involves policy support; if yes, it is 1; if no, it is 0
	X9.7 Social oversight	Whether the policy involves social oversight; if yes, it is 1; if no, it is 0
	X9.8 Technological innovation	Whether the policy involves technological innovation; if yes, it is 1; if no, it is 0
X10		Whether the policy is open and transparent; if yes, it is 1; if no, it is 0

Table 3 shows the evaluation criteria of second-level variables. If the policy text conforms to one of the second-level variables, it is coded as 1; otherwise, it is coded as 0.

**Table 4 ijerph-18-07676-t004:** Input–output policy table.

	X1				X2				X3		
	X1.1	X1.2	X1.3	X1.4	X2.1	X2.2	X2.3	X2.4	X3.1	X3.2	X3.3
P1	1	0	1	1	1	0	1	1	0	1	0
P2	1	1	1	1	1	0	1	1	1	0	0
P3	1	0	1	0	1	0	1	1	1	1	0
P4	1	0	1	1	1	0	1	1	1	0	0
	……				……				……		
P15	1	0	1	1	1	0	0	0	1	0	1
P16	1	1	0	0	0	0	1	1	1	0	1
……											
	**X6**			**X7**		**X8**					
P1	1	1	1	0	1	0	0	1	1	0	1
P2	1	1	1	0	1	0	0	1	1	1	1
P3	0	1	1	0	1	0	0	1	1	1	1
P4	0	1	0	0	1	1	1	1	0	0	1
	……				……				……		
P15	1	1	1	0	1	1	1	0	0	0	0
P16	1	1	0	0	1	0	1	1	0	0	1
……											

**Table 5 ijerph-18-07676-t005:** Policy PMC-Index and type.

	X1	X2	X3	X4	X5	X6	X7	X8	X9	X10	PMC-Index	Type	Rank
P1	0.75	0.75	0.33	1.00	0.83	1.00	0.50	0.50	0.38	1.00	7.04	Excellent	8
P2	1.00	0.75	0.33	1.00	0.50	1.00	0.50	0.67	0.63	1.00	7.38	Excellent	3
……													
P11	0.75	1.00	0.67	1.00	0.50	1.00	0.50	0.67	0.75	1.00	7.83	Excellent	1
P12	0.75	0.75	0.33	0.80	0.67	1.00	0.50	0.83	0.50	1.00	7.13	Excellent	6
P13	0.50	0.25	0.33	1.00	0.50	0.33	0.50	0.50	0.38	1.00	5.29	General	16
P14	1.00	0.75	0.33	1.00	0.50	1.00	0.50	0.33	0.88	1.00	7.29	Excellent	4
P15	0.75	0.25	0.67	1.00	0.50	1.00	0.50	0.33	0.75	1.00	6.75	General	11
P16	0.50	0.50	0.67	1.00	0.83	0.67	0.50	0.50	0.33	1.00	6.50	General	12
Total	11.00	11.25	7.33	15.40	9.00	12.33	8.00	9.17	9.83	16.00	--	--	--
Average	0.69	0.70	0.46	0.96	0.56	0.77	0.50	0.57	0.61	1.00	--	--	--

## Data Availability

The datasets used or analyzed during the current study are available from the corresponding author on reasonable request.

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
