# Peer review of "How Effective Is the Green Development Policy of China’s Yangtze River Economic Belt? A Quantitative Evaluation Based on the PMC-Index Model"

_ijerph, 2021, doi:10.3390/ijerph18147676_

Round 1
Reviewer 1 Report
OVERALL: Good paper; however, it needs some changes and would be a good contribution.
1. INTRODUCTION:
1.1. I think the introduction is well structured. However, it is not clear how the research question is structured, and it is also not clear how will be addressed. Also, I would clarify if there any similar case in Chine or elsewhere that could be used as the basis.
1.2. I don't understand the purpose of this sentence: "By evaluating the green development policy of the Yangtze River Economic Belt, we 62can identify problems in a policy text from the perspective of contradiction." I think it's broken.
1.3. Also, suddenly, you select Policy PMC-index as the primary research method without introducing it to the reader. I would suggest a paragraph explain it and why it is appropriate for this case study.
2. LIT REVIEW:
2.1 Now, you go back to the statement on point 1.2; this should be in the introduction. Explain why it is a contradiction.
2.2. You have many statements without any proper citation, such as "China's understanding of environmental issues has come later than western countries."
2.3 What late start are you referring to here (Line 102-103): "Given the late start of academic research on green development policy, there are few research results" really? I think that there is a lot of research about this topic.
3. Research Design:
3.1 I think that the part where you define PMC-Index should be in Lit Review, and go ahead and explain how you will apply it in this case.
4. Results
4.1 I get lost when you present table 4. It would help if you told the reader what I should be looking at. What is the main conclusion that I can be drawn from the results? Also, it applies to Fig 2. - Fig 7.
4.2 Point 4.2 should be in the Research Design.
5. - Conclusions
5.1 I got lost; there is no policy implication, although it evaluates a policy. I would encourage you to answer why I did this? Just an economic exercise or to suggest changes in current policies. That should be an important aspect of the paper.
Reviewer 2 Report
The review is attached.

Round 2
Reviewer 1 Report
Thanks for your response. Now, I think that the paper is ready to be accepted for publication. Congratulations!